# Novel Spinel Li–Cr Nano-Ferrites: Structure, Morphology, and Electrical/Dielectric Properties

**DOI:** 10.3390/ijms262110409

**Published:** 2025-10-26

**Authors:** Mukhametkali Mataev, Altynai Madiyarova, Moldir Abdraimova, Marzhan Nurbekova, Karima Seibekova, Zhanar Tursyn, Assel Kezdykbayeva, Krishnamoorthy Ramachandran, Bahadir Keskin

**Affiliations:** 1Department of Chemistry, Faculty of Natural Sciences, Kazakh State Women’s Teacher Training University, Almaty 050000, Kazakhstan; mataev.m@qyzpu.edu.kz (M.M.); nurbekova.m@qyzpu.edu.kz (M.N.); seitbekova.k@qyzpu.edu.kz (K.S.); tursyn.zh@qyzpu.edu.kz (Z.T.); 2Departments of Inorganic and Technical Chemistry, Faculty of Chemistry, E.A. Buketov Karaganda State University, Karagandy 100024, Kazakhstan; kezdikbayeva_assel@buketov.edu.kz; 3Department of Physics, SRM Institute of Science and Technology, Vadapalani Campus, Chennai 600026, India; ramachak1@srmist.edu.in; 4Department of Chemistry, Faculty of Arts & Science, Yildiz Technical University, TR34220 Istanbul, Turkey; bahadirkeskin@gmail.com

**Keywords:** sol-gel method, ferrites, symmetry, X-ray diffraction, pycnometric density, unit cell parameters, scanning electron microscope

## Abstract

This article reports on the synthesis and physicochemical characterization of a novel complex ferrite material, LiCr_3_._4_Fe_1_._6_O_8_, prepared via the sol-gel method. X-ray diffraction (XRD) analysis confirmed that the synthesized compound is a single-phase material with a spinel-type structure and cubic symmetry. Raman spectroscopy was employed to investigate the vibrational modes, and the observed peaks corresponding to Fe–O and Cr–O bonds further validated the spinel-like structure of the compound. The microstructure and elemental composition were examined using scanning electron microscopy (SEM). Multiple regions of the LiCr_3_._4_Fe_1_._6_O_8_ crystals were analyzed, revealing a homogeneous phase and providing detailed insight into the morphology and chemical composition of the surface. The synthesized ferrite particles exhibited relatively large dimensions, with sizes measured at approximately 5, 30, 100, and 200 μm. The dielectric behavior was studied to assess the material’s response to an external electric field, demonstrating its capacity for electric charge polarization. Both capacitance and electrical conductivity were found to increase with rising temperature. Electrophysical measurements were conducted using the LCR-800 system over a temperature range of 293–483 K and at frequencies of 1.5 kHz and 10 kHz. An increase in frequency to 10 kHz resulted in a decrease in the dielectric constant (ε) across the entire temperature range.

## 1. Introduction

Despite the rapid advancement in the field of materials science in recent years, there are still many unresolved issues. These include materials with perovskite, spinel, ferrite-garnet, and garnet-type structures [1]. Various methods are employed for obtaining multiferroic materials, including sol–gel, high-temperature solid-state synthesis, hot soap, spray pyrolysis, co-precipitation, and combustion techniques [2,3,4,5,6,7,8,9,10,11,12]. Among them, nano-sized ferrite particles are of particular interest due to their unique properties and wide range of applications. Ferrite materials have been widely used in numerous technological applications, including magnetically controlled drug delivery for cancer therapy, color imaging, gas sensing materials, and catalytic materials [13,14,15,16,17,18,19,20].

Spinel ferrite magnetic ceramics are gaining increasing attention in the electronics industry due to their exceptional properties, including high magnetic permeability, low dielectric losses, moderate saturation magnetization, strong magnetic induction, mechanical hardness, low eddy current losses, and ease of processing [21,22,23,24,25,26]. The demand for ferrites continues to grow rapidly, driven by their wide range of applications in fields such as biomedicine, targeted drug delivery, electromagnetic wave absorption, catalysis, computer memory systems, magnetic recording, and switching devices [27,28].

Among these, lithium (Li) ferrite stands out as a highly promising material for microwave technologies and is extensively used in electronic and communication systems [29]. Lithium ferrites are regarded as some of the most versatile ferrite materials due to their wide application spectrum. Owing to their excellent rectangular hysteresis loops, they are commonly utilized in data storage systems, switching components, and phase shifters [30,31,32]. With a relatively high Curie temperature of around 640 °C, lithium ferrites can operate effectively across a broad range of technically important temperatures. At room temperature, lithium ferrites offer better conductivity and higher saturation magnetization than yttrium iron garnet, while also being more cost-efficient. Moreover, they are widely used in lithium-ion batteries [33,34]. Lithium ferrite (Li_0_._5_Fe_2_._5_O_4_) is also recognized as a valuable material for use in construction and engineering sectors [35].

Among the various synthesis methods, the sol–gel technique is considered particularly effective and straightforward, as it allows for material formation at relatively low agglomeration temperatures. This method offers several benefits, including the use of low-cost precursors, operational simplicity, energy efficiency, and the ability to produce nanoparticles with fine, controlled structures.

The primary aim of this study is to explore the potential of lithium ferrites (Li_0_._5_Fe_2_._5_O_4_) in a variety of applications, including energy storage electrodes (such as those used in lithium-ion batteries), magnetic core inductors, military stealth technologies, and multilayer components in electronic devices. Accordingly, the investigation of synthesis parameters, X-ray structural analysis, and studies of the thermodynamic, physicochemical, and magnetic properties of these materials remain areas of significant scientific interest. This work presents a comprehensive investigation into the synthesis and characterization of ferrite nanoparticles using various preparation techniques.

## 2. Results

### 2.1. X-Ray Diffraction (XRD) Analysis

The XRD pattern of the synthesized LiCr_3_._4_Fe_1_._6_O_8_ complex ferrite was recorded at room temperature. As illustrated in Figure 1, the diffraction results confirm the complete transformation of the sample from an amorphous to a fully polycrystalline state, with the formation of a well-defined single phase. All observed peaks correspond to specific crystallographic planes, indicating the presence of a spinel-type cubic structure with space group $F\bar{d}3m$.

The X-ray analysis also showed that the lattice parameter of the LiCr_3_._4_Fe_1_._6_O_8_ ferrite is influenced by the calcination conditions. The absence of secondary or impurity phases in the diffraction pattern attests to the high purity and structural quality of the synthesized material. The X-ray pattern card number is No. 04-010-5163.

The figure above depicts the unit cell of the crystal structure. A unit cell represents the smallest repeating structural element in a crystalline material. In this case, the cell exhibits a cubic geometry, reflecting the material’s high degree of symmetry. Lithium (Li) and Chromium (Cr) atoms are shown in the same color, indicating that they are statistically distributed over identical crystallographic sites.

Lithium atoms are positioned at the tetrahedral sites, each surrounded by four oxygen atoms, while Chromium and Iron (Fe) atoms occupy octahedral sites, where they are coordinated by six oxygen atoms. The lines connecting the atoms represent the chemical bonds that maintain the integrity of the structure, which are predominantly ionic in nature.

The $F\bar{d}3m$ space group is a defining feature of spinel structures. Figure 2 shows the crystal structure of the complex ferrite LiCr_3_._4_Fe_1_._6_O_8_, which belongs to the $F\bar{d}3m$ space group. In the case of the LiCr_3_._4_Fe_1_._6_O_8_ compound, this crystallographic symmetry significantly influences its physical and chemical properties by determining the precise atomic arrangement and electronic structure. Such a configuration also has a pronounced effect on the magnetic properties of the material.

Below, the results of the Rietveld refinement based on the X-ray diffraction data for the synthesized complex ferrite are presented.

The reliability of the structural analysis is supported by the close agreement between the X-ray density and the values obtained from pycnometric measurements.

Elevated temperatures can contribute to the removal of structural defects or residual impurities, thereby enhancing crystal quality and promoting the growth of larger crystallites. The average crystallite size was estimated using the Debye–Scherrer equation, which is commonly used to determine crystallite dimensions from X-ray diffraction peak broadening. The Scherrer formula used for this calculation is as follows:(1)$$d=\frac{K\lambda }{\beta \cos\theta }$$
where

d—average size of crystals;

K—dimensionless particle shape coefficient (Scherrer constant);

λ—wavelength of X-ray radiation;

β—half-height reflex width (2θ in radians and units);

θ—diffraction angle (Bragg angle).

The microstructural characteristics, including crystallite size and microstrain, of the synthesized complex ferrite were determined using the Williamson–Hall (W–H) method. The Williamson–Hall equation is a widely used analytical approach for examining crystal structures and their mechanical properties. It enables the evaluation of microstructural features—specifically, the average crystallite size and lattice strain—based on data derived from X-ray diffraction analysis. The W–H equation is expressed as follows:(2)$$\beta cos\;\theta =\;\frac{\kappa \lambda }{D}+4\varepsilon sin\;\theta$$
where:

*β*—Full width of the diffraction peak (also called “width”) in radians,

*θ*—Diffraction angle,

*k*—A system factor required for the calculation (usually taken as 0.9),

*λ*—Wavelength of the X-ray radiation,

*D*—Average crystallite size (sometimes referred to as “crystal size”),

*ϵ*—Effect of internal strain, i.e., the expansion and compression due to stresses within the crystal.

To determine these parameters, a plot was constructed based on the relationship between βcos θ and 4sin θ, as illustrated in Figure 3. By applying linear regression to this plot, the slope of the line provides the microstrain, while the y-intercept corresponds to the crystallite size, as summarized in Table 1. The analysis was conducted using the Uniform Deformation Model (UDM), which assumes uniform strain in all crystallographic directions, indicating that the material exhibits isotropic behavior.

The crystallite sizes calculated using the Williamson–Hall method were slightly larger than those obtained from the Scherrer equation. This difference is primarily attributed to lattice strain induced by varying heat treatment temperatures [36,37,38,39,40]. As shown in Table 2, increasing the calcination temperature results in a decrease in lattice strain, which facilitates recrystallization and grain growth, thereby reducing structural defects and internal stresses.

### 2.2. Raman Spectroscopy

Raman spectroscopy was employed to analyze the phase structure and composition of the modified samples. This technique identifies the vibrational modes of molecules through molecular spectroscopy.

The measurements were performed using a SPECTRUM TERS system. A solid-state laser with a wavelength of 473 nm and a maximum power output of 35 kW was used for excitation, focusing on a laser spot of approximately 2 μm in diameter on the sample surface. The Raman signal was collected in backscattering geometry using an SSD detector, while the sample was cooled to −65 °C. The spectral resolution of the 600/600 grating was 4 cm^−1^, with a signal accumulation time of 30 s.

The *Y*-axis represents normalized intensity, enabling the comparison of relative Raman signal strengths across different spectra. The *X*-axis indicates the Raman shift in cm^−1^, which corresponds to the vibrational energy levels of the molecules within the sample.

Figure 4 shows three Raman spectra. Peaks near the 200 cm^−1^ region (at 195, 200, and 206 cm^−1^) are attributed to lattice vibrations involving Li^+^, Cr^3+^, or Fe^3+^ cations. In the 500–600 cm^−1^ range (509, 511, 516, 597 cm^−1^), the peaks correspond to Fe–O bond vibrations within the spinel lattice. Peaks observed between 600 and 700 cm^−1^ (at 608, 618, 650, 690, and 705 cm^−1^) relate to Fe–O vibrational modes in ferrite groups, especially those involving CrO_4_^2−^ units. A prominent peak around 720 cm^−1^ signals a high concentration of ferrite groups.

The detection of characteristic Fe–O and Cr–O vibrational peaks confirms that the LiCr_3_._4_Fe_1_._6_O_8_ sample has a spinel-type crystal structure. Variations in peak intensity and position among the three spectra suggest sample heterogeneity or differences in crystallite orientation in different regions. In addition, the colors (black, red, and blue lines) generally represent LiCr_3_._4_Fe_1_._6_O_8_ ferrite samples obtained at different calcination temperatures. The blue line corresponds to the sample synthesized at a lower temperature, the red line represents the sample prepared at a medium temperature, and the black line indicates the sample obtained at a higher temperature.

As the temperature increases, the shift and intensity of the peaks change—this indicates the improvement of the crystalline structure, the strengthening of ionic bonds, or possible phase transitions.

Overall, the Raman spectrum of the LiCr_3_._4_Fe_1_._6_O_8_ sample indicates a complex phase composition, consisting of both spinel structures and ferrite groups.

Raman (or combination scattering) spectroscopy provides a graphical representation of light scattering resulting from its interaction with molecules in a material. While Rayleigh scattering occurs without an energy change, Raman scattering involves a shift in energy corresponding to molecular vibrational modes. This scattering delivers important insights into the chemical composition and structural properties of the material.

As shown in Figure 5, the X-axis (Raman shift, cm^−1^) represents the change in scattered light energy measured in reciprocal centimeters (cm^−1^). Each vertex along this axis corresponds to a specific molecular oscillation mode in the sample.

The *Y*-axis (Intensity, counts) indicates the intensity of the scattered light, which reflects the number of molecules participating in a particular vibrational mode. Higher peak intensities correspond to a larger number of molecules vibrating in that mode.

The spectrum displays several peaks with varying intensities, each representing distinct vibrational modes of the sample’s molecules. For instance, the peak at 851 cm^−1^ is the most intense, signifying a dominant vibrational mode. Peaks at 892 cm^−1^ and 930 cm^−1^, situated near the 851 cm^−1^ peak, correspond to other vibrational modes or different components within the sample. Peaks at 348 cm^−1^ and 400 cm^−1^ have lower intensities, suggesting they arise from less common vibrational modes or impurities present in the sample.

### 2.3. Scanning Electron Microscope

A scanning electron microscope (SEM) (APPLICATION Note Team, Bruker, Berlin, Germany) was utilized to analyze the distribution spectrum, perform both qualitative and quantitative assessments, and determine the elemental composition by percentage.

To examine the surface morphology of the newly synthesized complex mixed ferrite LiCr_3_._4_Fe_1_._6_O_8_ samples prepared via the sol-gel method, an electron microscope capable of scanning electric diffraction images and illuminating microstructures was employed. Various parameters were calculated, all of which aligned well with the X-ray analysis results. The presence of dark regions behind the nanostructures suggests the absence of nanomaterials in those areas. Furthermore, the micrographs reveal well-formed single-phase spinel ferrites, which are ideal for applications such as microwave absorption. Electron microscopy images of the compound’s surface are shown in Figure 6a–d.

The micrographs taken at magnifications of 5, 30, 100, and 200 μm are presented above, showing the overall morphology of the surface layer of the complex ferrite. The analysis confirmed that the compound is single-phase, with its structural clarity defined by its topography and chemical composition. Figure 7 displays the spectral data and elemental analysis results for the synthesized new mixed complex ferrite LiCr_3_._4_Fe_1_._6_O_8_.

The microstructure, chemical composition, and elemental distribution of chromium, iron, lithium, and oxygen within the sample were investigated based on the elemental mapping and crystallization characteristics. Quantitative elemental analysis reveals that iron, chromium, lithium, oxygen, and carbon atoms are distributed across regions measuring approximately 100 μm. Scanning electron microscopy also allowed observation of nanometer-sized particles within the powdered solid material.

The elemental distribution, along with quantitative and qualitative analyses, was performed using scanning electron microscopy. Figure 8 presents the spectral data of the synthesized new complex ferrite and the corresponding elemental analysis results.

The elemental composition and distribution of the synthesized complex ferrite LiCr_3_._4_Fe_1_._6_O_8_ were investigated using energy-dispersive X-ray (EDX) spectroscopy. The obtained EDX spectrum confirmed the presence of the main elements—lithium (Li), chromium (Cr), iron (Fe), and oxygen (O)—consistent with the theoretical stoichiometric composition of the compound.

Quantitative analysis revealed slight deviations in the relative atomic percentages of the elements compared to the theoretical values. These differences can be attributed to the surface sensitivity of the EDX technique and the inherent difficulty in accurately detecting light elements, particularly lithium.

A comparison of theoretical and experimental data showed that the elements are uniformly distributed according to the stoichiometric ratio of the LiCr_3_._4_Fe_1_._6_O_8_ phase, confirming the homogeneous structure of the synthesized material. In addition, the slightly higher amounts of oxygen and lithium observed in the spectrum may be associated with surface oxidation or lithium accumulation at grain boundaries.

Overall, the EDX analysis confirms that the obtained ferrite forms a single-phase spinel structure with a uniform distribution of elements.

### 2.4. Broad Band Spectrometer Analysis

The variation of the capacitance (Cp) of LiCr_3_._4_Fe_1_._6_O_8_ ferrite with frequency at different temperatures is shown in Figure 9. At low frequencies, the capacitance exhibits relatively high values, while it gradually decreases as the frequency increases. This behavior can be attributed to space charge and interfacial polarization effects that dominate at low frequencies, whereas at higher frequencies, the dipoles fail to follow the alternating field, leading to a reduction in polarization. Furthermore, the increase in temperature enhances the mobility of charge carriers and the degree of polarization up to a certain limit, after which further heating results in a decrease in capacitance due to thermal vibrations and structural relaxation effects.

According to the obtained results, the capacitance properties of the complex LiCr_3_._4_Fe_1_._6_O_8_ ferrite are strongly temperature-dependent. With increasing temperature, the capacitance (Cp) of the material gradually rises and remains nearly constant up to approximately 303 °C. In this range, the increase in the thermal mobility of charge carriers and the strengthening of dipolar polarization play a key role. However, beyond 303 °C, a sharp decline in capacitance is observed. This phenomenon is associated with a reduction in defect concentration within the crystal lattice, the weakening of intergranular barrier layers, and changes in the relaxation dynamics of charge carriers under high-temperature conditions.

Overall, the obtained results indicate that the dielectric properties of LiCr_3_._4_Fe_1_._6_O_8_ ferrite are highly sensitive to its structural stability and temperature effects. This suggests that the material holds potential for application in electronic and energy-storage devices operating in high-temperature environments.

The real part of the modulus increases similarly with frequency. The presence of two plateau regions indicates the existence of two distinct relaxation processes in the material, which are reflected by two peaks in the modulus (M) curve. The peak observed in the low-frequency region corresponds to the long-range hopping motion of charge carriers, where charges move from one ionic site to neighboring sites at the grain boundaries. In contrast, the peak appearing in the high-frequency region (observed at 313 K) is associated with short-range hopping motion, where ions are spatially confined and exhibit localized movement within the grains. As the temperature increases, the shift of this peak toward higher frequencies, as shown in Figure 10, suggests that the short-range hopping transport process becomes more dominant at elevated temperatures.

Figure 11 shows that the conductivity of the samples is nearly independent of frequency in the low-frequency region. Their high insulating nature, characterized by low conductivity values ranging from 10^2^ to 10^6^ S/cm, and the stability of conductivity over a wide range of low frequencies make these samples suitable materials for the fabrication of high-quality resistors.

In a wide range of ceramic materials, conduction can occur through ionic conduction, mixed ionic–electronic conduction, or purely electronic conduction, while grain boundaries act as barriers to the movement of charge carriers. This barrier effect is generally more pronounced at lower temperatures (Figure 12). In this temperature range, perovskite-type titanates such as LiCr_3_._4_Fe_1_._6_O_8_ are widely used in capacitor applications as materials with high dielectric permittivity. At temperatures well above the Curie point, LiCr_3_._4_Fe_1_._6_O_8_-based ceramics exhibit a pronounced effect of highly resistive grain boundaries (GB). This phenomenon is attributed to the space charge effect of a Schottky-type depletion layer. In such applications, titanates must be acceptor-doped to prevent semiconducting behavior in the ferrite materials.

Figure 13 shows the dielectric loss of the material, which is highly sensitive to both frequency and temperature. The shift of the dielectric relaxation peaks toward higher frequencies with increasing temperature clearly indicates that this process is thermally activated.

At low frequencies, the highly resistive grain boundaries exhibit reactive behavior, leading to greater absorption of electrical energy as current flows through the material. The variation in specific resistance with temperature for LiCr_3_._4_Fe_1_._6_O_8_ across frequencies ranging from 1 Hz to 1 MHz is shown above. This behavior is attributed to the fact that increasing temperature enhances the mobility of charge carriers, while the applied electric field acts as a driving force, aligning the electric dipoles in its direction. The data indicate that specific resistance decreases with increasing temperature and frequency.

### 2.5. Electrophysical Research

The procedure followed the established methods for measuring electrophysical properties [41,42]. The dielectric constant and electrical resistance were investigated by measuring the electrical capacitance of the samples using an LCR-800 serial device (Taiwan) under thermostatic conditions at a working frequency of 1 kHz in a continuous dry air environment, with a holding time at each specified temperature.

The samples were prepared as disc-shaped pellets, parallel to the plane, with a diameter of 10 mm and a thickness ranging from 2 to 6 mm. A binder mixture (1.5%) was added prior to pressing, which was performed at a pressure of 20 kg/cm^2^. The resulting discs were sintered in a SNOL laboratory furnace at 400 °C for 6 h, followed by careful surface leveling on both sides.

The dielectric constant was calculated based on the sample’s capacitance, taking into account its thickness and the surface area of the electrodes. A Sawyer–Tower circuit was employed to determine the relationship between electric displacement *D* and electric field strength *E*. The hysteresis loop *D* was visually monitored using a C1-83 oscilloscope, connected to a voltage divider consisting of resistances of 6 MΩ and 700 kΩ, along with a reference capacitor of 0.15 µF. The generator frequency was set to 300 Hz.

During all temperature-dependent measurements, the samples were placed inside a furnace. Temperature was monitored using a chromel–alumel (chromium–aluminum) thermocouple connected to a B2-34 voltmeter, with a measurement error of ±0.1 mV. The heating rate was maintained at 5 K/min. The dielectric constant at each temperature was calculated using the following formula:(3)$$\varepsilon =\frac{C}{C_{0}}$$
where *C*_0_ is the capacitance of the capacitor without the test material (in air).

The band gap energy (ΔE) of the material under investigation was calculated using the following formula:$$∆E=\frac{2kT_{1}T_{2}}{0.43(T_{2}-T_{1})}log\frac{R_{1}}{R_{2}}$$where k is the Boltzmann constant equal to 8.6173303 × 10^−5^ eB·K^−1^; R_1_ is the resistance at T_1_; and R_2_ is the resistance at T_2_.

To verify the accuracy of the obtained data, the dielectric constant of a standard reference material, barium titanate (BaTiO_3_), was measured at frequencies of 1 kHz, 5 kHz, and 10 kHz. Table 3 presents the measured electrophysical properties of BaTiO_3_.

As shown in Table 3, the dielectric constant values of BaTiO_3_ at 293 K for frequencies of 1 kHz and 5 kHz are in good agreement with the recommended value of 1400 ± 250. Furthermore, the observed changes in the electrical conductivity of BaTiO_3_ at all measured frequencies (1 kHz, 5 kHz, and 10 kHz) at 383 K correspond to its phase transition from the cubic perovskite structure (space group Pm3m) to the tetragonal (polar) ferroelectric phase (space group P4mm) [43,44,45].

It is worth noting that although a decrease in the dielectric constant of BaTiO_3_ was recorded at 10 kHz and temperatures of 293 K, 303 K, and 313 K, the dielectric constant (ε) values across the full temperature range of 313–483 K remain as high as approximately 2150 at all three frequencies (1 kHz, 5 kHz, and 10 kHz). This indicates that frequency variation within this range has little impact on the temperature dependence of the dielectric constant for BaTiO_3_.

Electrophysical measurements for LiCr_3_._4_Fe_1_._6_O_8_ were performed using the LCR setup over the temperature range of 293–483 K at frequencies of 1, 5, and 10 kHz (see Table 4 and Figure 14).

The data presented in Table 4 and Table 5 and Figure 14 indicate that the dielectric permittivity (ε) of material 1 at 293 K is relatively high, measured at 2.69 × 10^5^, and it reaches a peak value of 2.47 × 10^6^ at 373 K. As the temperature further increases to the range of 423–483 K, the ε values decrease significantly, dropping to 122 at 483 K. Additionally, increasing the frequency to 10 kHz causes a reduction in ε throughout the entire temperature range (293–483 K).

The temperature dependence of electrical resistance reveals complex conductivity behavior: from 293 K to 313 K, the material behaves like a semiconductor; between 313 K and 343 K, it exhibits metallic conduction; from 343 K to 363 K, it returns to semiconductor-like behavior; and from 363 K to 483 K, it again shows metallic conductivity.$$\Delta E=\frac{2\times 0.00086173\times 293\times 313}{0.43(313-293)}\log\frac{4.41}{4.22}=1.92\;эB$$$$\Delta E=\frac{2\times 0.000086173\times 343\times 363}{0.43(363-343)}\log\frac{6.10}{3.81}=4.0\;эB$$

## 3. Discussion

A comprehensive investigation of the structural and physical properties of the LiCr_3_._4_Fe_1_._6_O_8_ sample confirmed the formation of a well-defined cubic spinel structure. The average crystallite size was determined to be 5.2 nm using the Scherrer equation and 9.12 nm from the Williamson–Hall analysis, while the calculated microstrain (ε = 1.71×10^−3^) indicated lattice distortion associated with the incorporation of Cr^3+^ ions. Raman spectra exhibited characteristic vibrational modes: Li^+^, Cr^3+^, and Fe^3+^ ions near 200 cm^−1^, Fe–O stretching vibrations within 500–600 cm^−1^, and Cr–O vibrations in the 600–700 cm^−1^ range. A sharp band at approximately 720 cm^−1^ further confirmed the high degree of structural ordering in the ferrite phase. SEM micrographs revealed uniformly distributed nanostructured grains, while elemental mapping (Cr, Fe, Li, O) demonstrated a homogeneous distribution of the constituent elements, confirming the effectiveness of the synthesis process. Dielectric measurements showed that the dielectric constant (ε′) increased with rising temperature, reaching a maximum value of ~2.5×10^6^ in the range of 353–373 K. Conversely, ε′ decreased with increasing frequency, which can be attributed to Maxwell–Wagner interfacial polarization and hopping conduction of charge carriers. Furthermore, the dielectric loss (tan δ) exhibited a decreasing trend with frequency, indicating that the material displays stable and typical dielectric behavior.

## 4. Materials and Methods

The sol–gel method was utilized as an efficient technique for synthesizing a novel complex mixed ferrite with a heterogeneous composition. To identify the composition of the synthesized ferrite, X-ray (Osaka, Japan) phase analysis was performed. Scanning electron microscopy (SEM) (Waltham, MA, USA) was used for both qualitative and quantitative characterization of the material. In addition, the electrophysical properties, such as dielectric conductivity and electrical resistivity, were thoroughly investigated.

The crystal structure of lithium chromium ferrite powder samples was examined using X-ray diffraction (XRD) analysis. Phase identification was carried out with a Rigaku Miniflex/600 (Tokyo, Japan) diffractometer, operating with CuKα radiation (U = 30 kV, I = 10 mA), a scanning speed of 1000 pulses per second, and a time constant of 5 s. Data were collected over a 2θ range of 5° to 90° and analyzed using the PDF-5+ database.

The synthesis process began with high-purity reagent-grade chromium (III) oxide, iron (III) oxide, and lithium carbonate as the primary raw materials, along with distilled water. Glycerol (a trihydric alcohol) and citric acid served as the gel-forming agents. The stoichiometric amounts of the raw materials were precisely weighed to an accuracy of 0.0001 g using an analytical balance, followed by mixing and grinding in an agate mortar. The prepared mixture was then transferred to an alumina crucible and dried at 100 °C in an electric furnace to promote gel formation.

After drying, the gel was subjected to calcination at 600 °C in a muffle furnace for 6 h, resulting in the formation of a fine black powder. This powder was then sintered in air, with the temperature gradually increased from 700 °C to 1000 °C over a total duration of 30 h, using 6-h increments at each temperature step [46].

The pycnometric density of the ferrite samples was determined using the liquid displacement method [47], employing toluene and deionized water as inert displacement fluids. Each sample’s density was measured five times, and the average values were calculated to ensure accuracy.

## 5. Conclusions

In conclusion, the novel mixed complex ferrite with the composition LiCr_3_._4_Fe_1_._6_O_8_ was successfully synthesized for the first time using the sol-gel method. To characterize the new complex ferrite, X-ray phase analysis, scanning electron microscopy (for both qualitative and quantitative elemental analysis), and electrophysical property measurements (including dielectric constant and electrical resistance) were performed.

For the first time, X-ray phase analysis was used to determine the symmetry types and unit cell parameters of the synthesized complex ferrite. The compound LiCr_3_._4_Fe_1_._6_O_8_ was identified as cubic (a = b = c = 8.3001 Å, Z = 2) with calculated and pycnometric densities of 4.745 g/cm^3^ and 4.748 g/cm^3^, respectively. The X-ray results confirmed that the synthesized material is polycrystalline, and the crystallographic data accuracy is supported by the close agreement between X-ray and pycnometric densities.

Raman spectroscopy was employed to study the vibrational modes of the molecules. The presence of characteristic Fe–O and Cr–O vibrational peaks confirmed that the LiCr_3_._4_Fe_1_._6_O_8_ sample possesses a spinel-type structure. Variations in intensity and peak positions among the spectra suggest sample heterogeneity or differing crystal orientations in various regions. The Raman spectrum indicated a complex phase composition, including spinel structures and ferrite groups.

Scanning electron microscopy allowed the collection of micro-samples from different regions of the LiCr_3_._4_Fe_1_._6_O_8_ crystals for elemental composition analysis. The overall morphology of the complex ferrite’s surface was observed, confirming that the compound is single-phase. The structure was further elucidated by the topography and chemical composition analysis. The elemental analysis results, presented in tabular form, confirmed the formula LiCr_3_._4_Fe_1_._6_O_8_. The particle sizes ranged from 5 µm to 200 µm.

Electrophysical studies showed that the LiCr_3_._4_Fe_1_._6_O_8_ ferrite’s capacitance and conductivity increase with temperature. The material exhibits two distinct relaxation mechanisms depending on frequency: long-range hopping at low frequencies and short-range hopping at high frequencies. At low frequencies, the material exhibits low conductivity, making it suitable for resistor applications. Grain boundaries contribute high resistance at elevated temperatures, while dielectric losses decrease with increasing frequency. The real part of the resistance decreases with both temperature and frequency, indicating the material’s potential use in electronic devices.

Electrophysical measurements of LiCr_3_._4_Fe_1_._6_O_8_ were conducted over the temperature range of 293–483 K at frequencies of 1, 5, and 10 kHz. At 293 K and 1 kHz, the dielectric constant (ε) was 2.69 × 10^5^, reaching a maximum of 2.47 × 10^6^ at 373 K. Increasing the frequency to 10 kHz resulted in a decrease in ε across the entire temperature range studied.

The temperature dependence of electrical resistance revealed complex conductivity behavior: semiconductor-like conductivity from 293 to 313 K, metallic behavior between 313 and 343 K, semiconductor-like again from 343 to 363 K, and metallic conduction from 363 to 483 K. The band gap was measured to be 1.92 eV within the 293–313 K range, typical of narrow-gap semiconductors, while at 343–363 K, it increased to 4.0 eV, characteristic of wide-gap semiconductors.

Overall, the obtained results suggest that LiCr_3_._4_Fe_1_._6_O_8_ ferrite is a promising material for potential applications in electronics and energy storage devices.

## Figures and Tables

**Figure 1 ijms-26-10409-f001:**
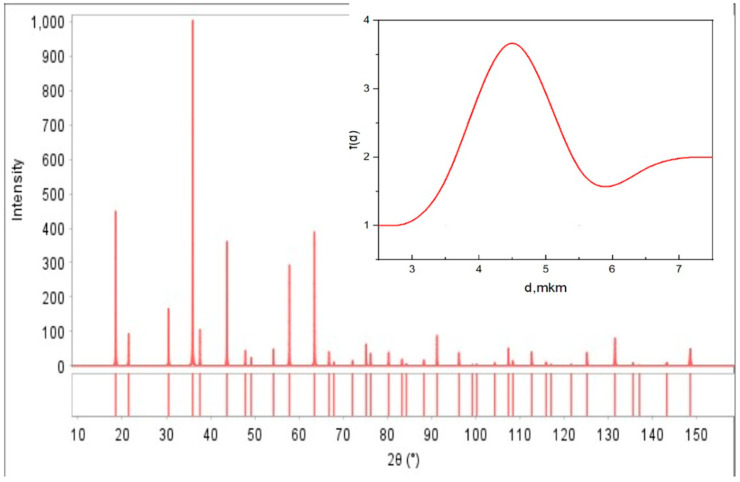
X-ray diffraction pattern of the complex ferrite LiCr_3_._4_Fe_1_._6_O_8_.

**Figure 2 ijms-26-10409-f002:**
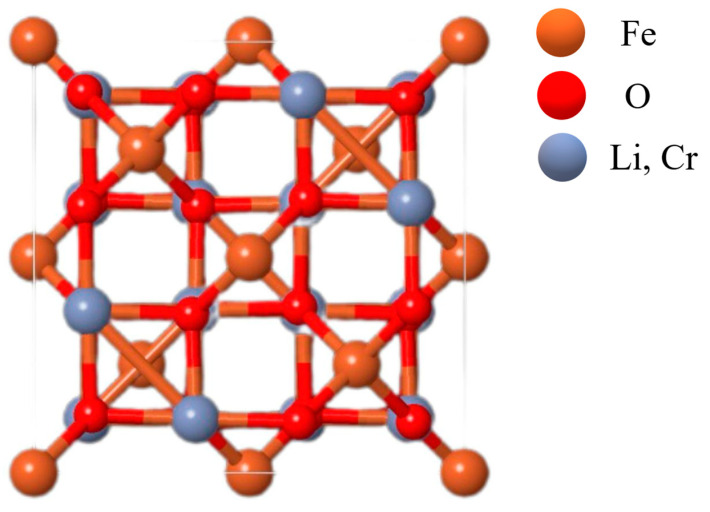
Crystal structure of the complex ferrite LiCr_3_._4_Fe_1_._6_O_8_ with space group $F\bar{d}3m$.

**Figure 3 ijms-26-10409-f003:**
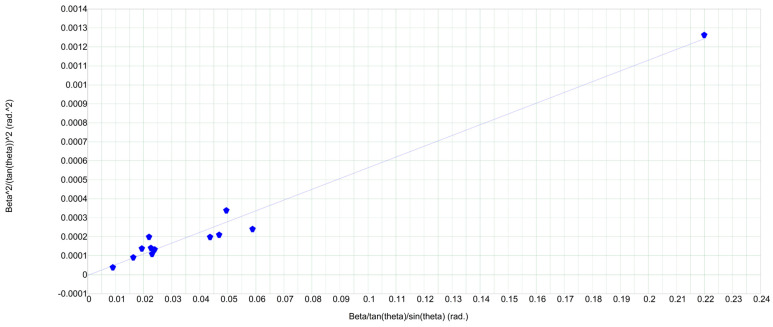
Williamson-Hall (W-H) plot of the LiCr_3_._4_Fe_1_._6_O_8_ complex ferrite.

**Figure 4 ijms-26-10409-f004:**
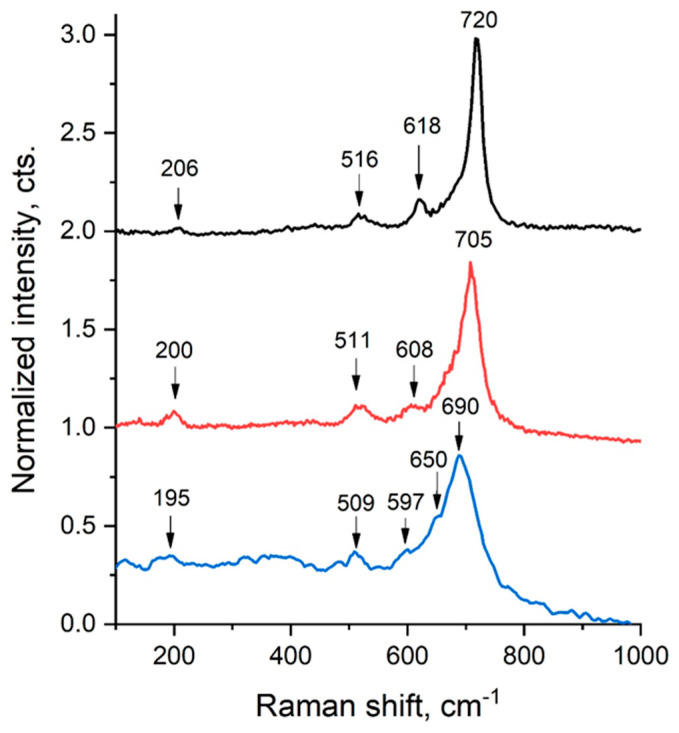
Raman Spectrum of the LiCr_3_._4_Fe_1_._6_O_8_ Sample.

**Figure 5 ijms-26-10409-f005:**
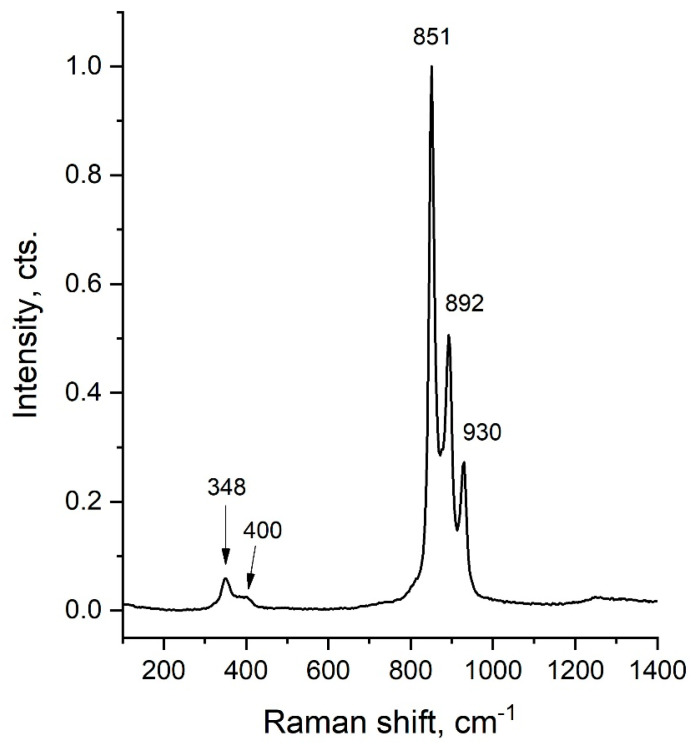
Spectrum of the sample (LiCr_3_._4_Fe_1_._6_O_8_) similar to the typical spinel structure signal.

**Figure 6 ijms-26-10409-f006:**
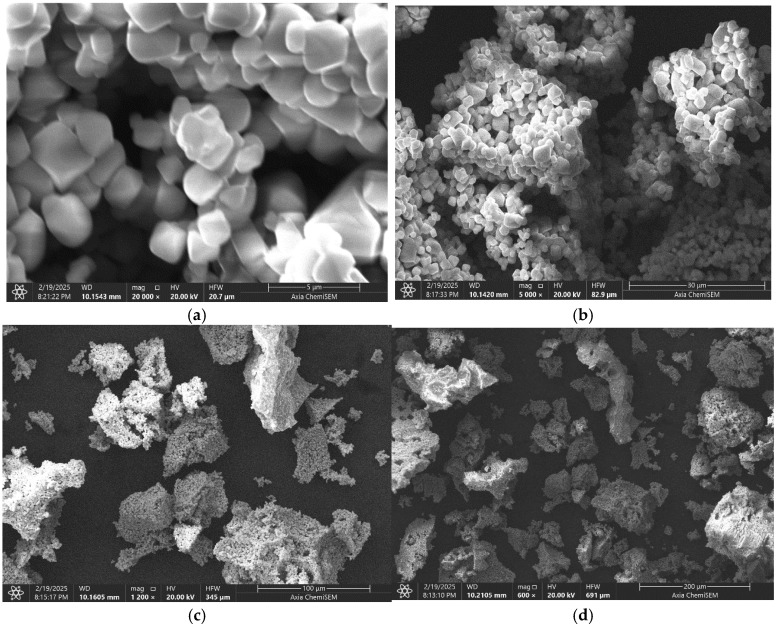
Micrometer-precision images of the new mixed complex ferrite LiCr_3_._4_Fe_1_._6_O_8_ captured at four different magnification scales.

**Figure 7 ijms-26-10409-f007:**
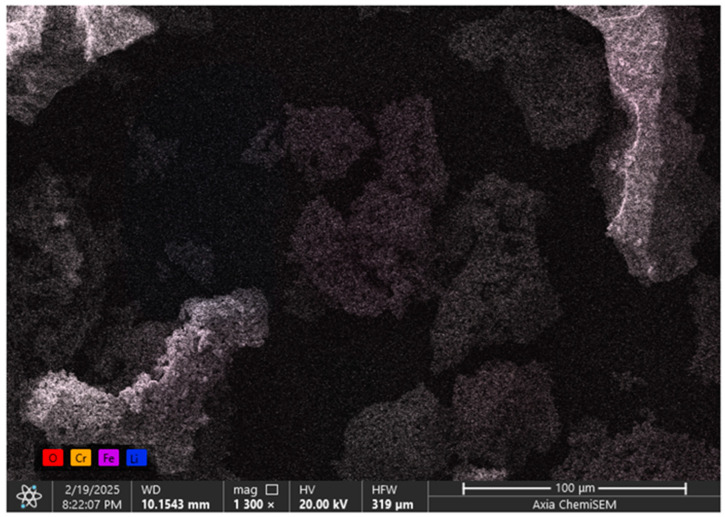
Distribution map of elements in the new mixed complex ferrite LiCr_3_._4_Fe_1_._6_O_8_ (distribution order and color coding of Cr, Fe, Li, O elements in the map).

**Figure 8 ijms-26-10409-f008:**
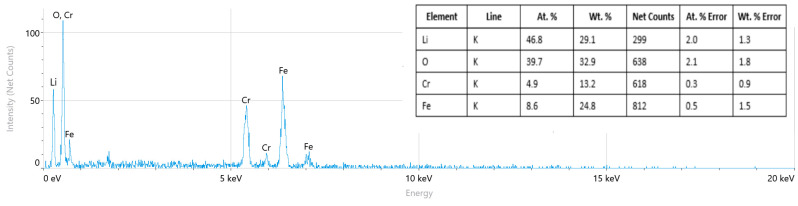
Spectral samples of the LiCr_3_._4_Fe_1_._6_O_8_ compound. Elemental analysis results are integrated.

**Figure 9 ijms-26-10409-f009:**
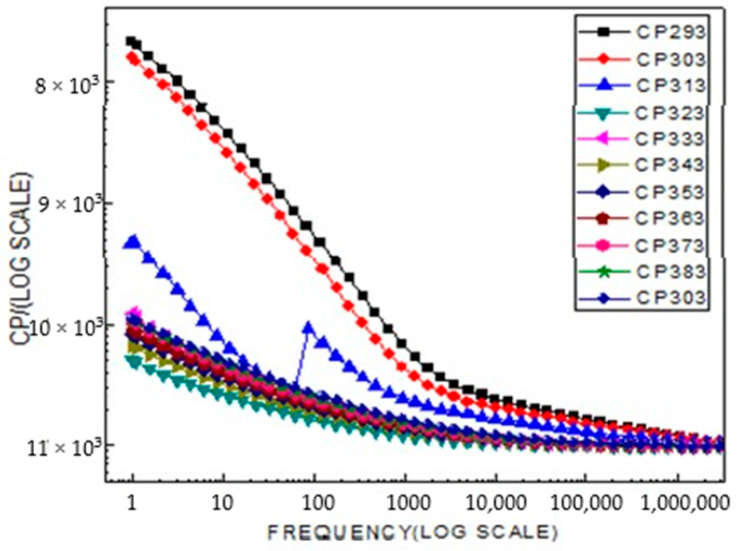
A part from Phil parameter, electrical conduction also increases with increase temperature.

**Figure 10 ijms-26-10409-f010:**
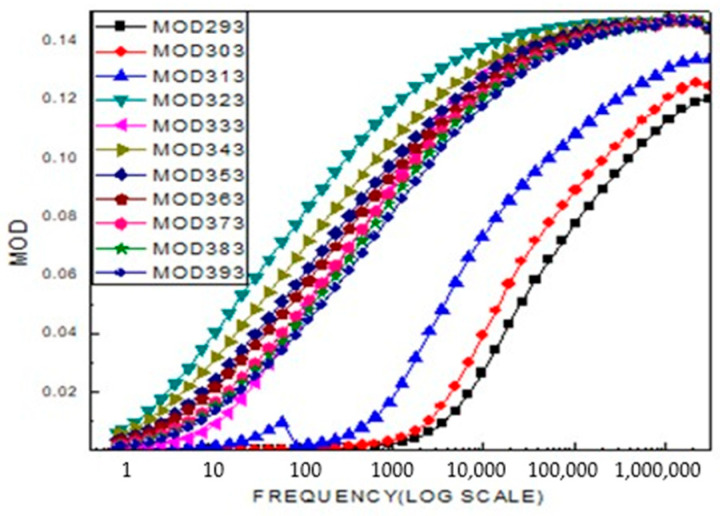
Shows a typical frequency dependent modulus real: M(LiCr_3_._4_Fe_1_._6_O_8_).

**Figure 11 ijms-26-10409-f011:**
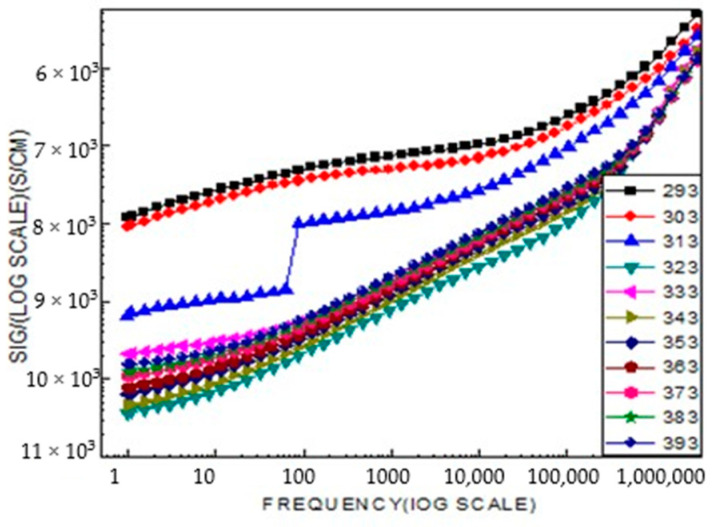
Shows the frequency (f) dependence of the real part σ of a.c. conductivity at selected temperatures in the range 293 K–393 K for the LiCr_3_._4_Fe_1_._6_O_8_ sample.

**Figure 12 ijms-26-10409-f012:**
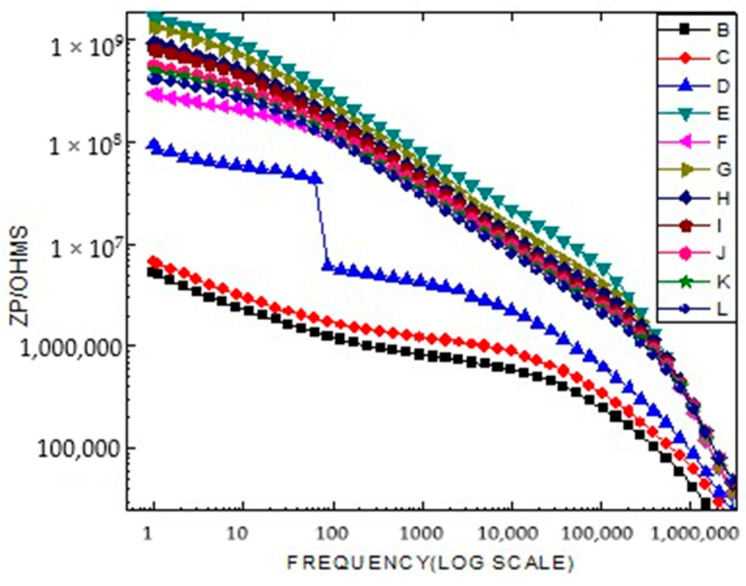
Grain boundaries play an important role in the electrical properties of a variety of ferrite materials and components.

**Figure 13 ijms-26-10409-f013:**
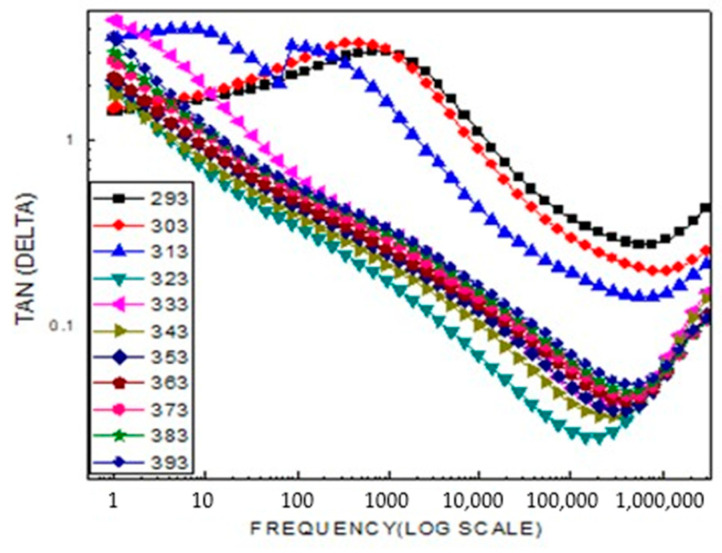
Dielectric loss (Tan δ) show that dielectric loss rapidly decreases with increasing frequency and fall to 1 above 100 Hz.

**Figure 14 ijms-26-10409-f014:**
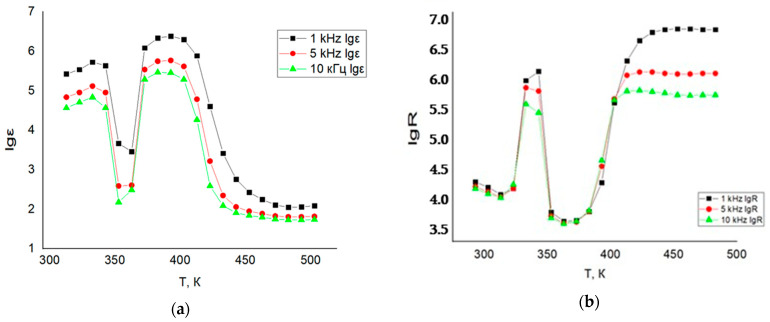
Temperature and frequency (1 kHz, 5 kHz, and 10 kHz) dependence of the dielectric permittivity (**a**) and electrical resistance (**b**) of LiCr_3_._4_Fe_1_._6_O_8_.

**Table 1 ijms-26-10409-t001:** This is a table. Structural parameters of LiCr_3_._4_Fe_1_._6_O_8_ obtained by Rietveld refinement of X-ray diffraction data.

Sample	LiCr_3.4_Fe_1.6_O_8_
Space Group	$F\bar{d}3m$, the cubic side is centered
Z	4
Cell Parameter (Å)	
a =	8279 Å
b = c = V(Å^3^)	8279 Å 8279 Å 57.646 Å^3^
Agreement factors R-factor I/Ic I/Ic- CW ND α β γ X-ray density (g/cm^3^) Pycn.density (g/cm^3^)	0.036 3.67 0.76 90 90 90 4.694 g/cm^3^ 4.695 g/cm^3^

**Table 2 ijms-26-10409-t002:** Average crystallite size and average strain for all ferrite samples fired at different temperatures, according to the Scherrer and W-H methods.

Sample	Scherrer Method	Williamson-Hall Method	
	D (nm)	D (nm)	ε × 10^−3^
LiCr_3.4_Fe_1.6_O_8_	5.2	9.12	1.71

**Table 3 ijms-26-10409-t003:** Dependence of electrical resistance (R), electrical capacity (C), and dielectric constant (ε) on BaTiO_3_ temperature.

T, K	C, nF	R, Oм	ε	lgε	lgR
Measurement frequency at 1 kHz					
293 303 313 323 333 343 353 363 373 383 393 403 413 423 433 443 453 463 473 483	0.27278 0.27426 0.27715 0.28125 0.28772 0.29313 0.29916 0.30751 0.31202 0.31702 0.32255 0.32967 0.3423 0.35119 0.36668 0.38018 0.39802 0.4169 0.43147 0.45456	13,400 13,270 12,910 12,560 11,890 11,210 10,290 9383 8831 9061 8814 7881 7098 6902 6153 6317 6010 5584 5149 4656	1296 1303 1316 1336 1367 1392 1421 1461 1482 1506 1532 1566 1626 1668 1742 1806 1891 1980 2050 2159	3.11 3.11 3.12 3.13 3.14 3.14 3.15 3.16 3.17 3.18 3.19 3.19 3.21 3.22 3.24 3.26 3.28 3.30 3.31 3.33	4.13 4.12 4.11 4.10 4.08 4.05 4.01 3.97 3.95 3.96 3.95 3.90 3.85 3.84 3.79 3.80 3.78 3.75 3.71 3.67
Measurement frequency at 5 kHz					
293 303 313 323 333 343 353 363 373 383 393 403 413 423 433 443 453 463 473 483	0.25678 0.2683 0.2775 0.28638 0.29667 0.30226 0.30787 0.31283 0.31843 0.32148 0.32578 0.32976 0.33303 0.33948 0.35613 0.3713 0.3925 0.41682 0.44245	29,630 21,650 13,080 5236 4301 4733 3296 2966 2805 2529 2669 3172 4434 6377 9644 11,520 10,430 8021 5978 3799	1220 1274 1318 1360 1389 1409 1436 1462 1486 1513 1527 1547 1566 1582 1613 1692 1764 1864 1980 2102	3.09 3.11 3.12 3.13 3.14 3.15 3.16 3.17 3.17 3.18 3.18 3.19 3.19 3.20 3.21 3.23 3.25 3.27 3.30 3.32	4.47 4.34 4.12 3.72 3.63 3.68 3.52 3.47 3.45 3.40 3.43 3.50 3.65 3.80 3.98 4.06 4.02 3.90 3.78 3.58
Measurement frequency at 10 kHz					
293 303 313 323 333 343 353 363 373 383 393 403 413 423 433 443 453 463 473 483	0.11814 0.18494 0.22927 0.25954 0.27501 0.28531 0.29302 0.29988 0.30652 0.31215 0.31667 0.32294 0.32779 0.33406 0.34256 0.35658 0.378 0.39475 0.41687 0.44203	152,300 70,790 32,200 11,870 4842 3312 2689 2257 1946 1689 1737 3130 5945 8231 8805 8052 5967 4604 3343 2353	561 878 1089 1233 1306 1355 1392 1424 1456 1483 1504 1534 1557 1587 1627 1694 1796 1875 1980 2100	2.75 2.94 3.04 3.09 3.12 3.13 3.14 3.15 3.16 3.17 3.18 3.19 3.19 3.20 3.21 3.23 3.25 3.27 3.30 3.32	5.18 4.85 4.51 4.07 3.69 3.52 3.43 3.35 3.29 3.23 3.24 3.50 3.77 3.92 3.94 3.91 3.78 3.66 3.52 3.37

**Table 4 ijms-26-10409-t004:** Dependence of electrical capacity (C), electrical resistance (R), and dielectric constant (ε) of LiCr_3_._4_Fe_1_._6_O_8_ on temperature and frequency.

T, K	C, nF	R, Oм	ε	Lgε	lgR
Measurement frequency at 1 kHz					
293 303 313 323 333 343 353 363 373 383 393 403 413 423 433 443 453 463 473 483	7.7977 10.302 15.68 12.489 8.1684 14.516 86.933 176.65 265.14 378.56 495.27 312.87 11.039 1.8105 0.95809 0.68585 0.70009 0.89737 1.3483 2.1329	165,700 135,800 101,700 131,900 203,600 121,200 32,510 19,300 14,110 10,520 8184 10,720 85,500 379,400 649,900 843,600 874,900 796,100 666,100 511,600	22,448 29,658 45,140 35,954 23,515 41,789 250,266 508,547 763,295 1,089,813 1,425,802 900,702 31,779 5212 2758 1974 2015 2583 3882 6140	4.35 4.47 4.65 4.56 4.37 4.62 5.40 5.71 5.88 6.04 6.15 5.95 4.50 3.72 3.44 3.30 3.30 3.41 3.59 3.79	5.22 5.13 5.01 5.12 5.31 5.08 4.51 4.29 4.15 4.02 3.91 4.03 4.93 5.58 5.81 5.93 5.94 5.90 5.82 5.71
Measurement frequency at 5 kHz					
293 303 313 323 333 343 353 363 373 383 393 403 413 423 433 443 453 463 473 483	1.3926 1.8886 2.728 1.517 0.9064 2.5719 15.47 31.031 45.909 64.608 82.944 40.501 1.7403 0.28075 0.13233 0.09429 0.09084 0.10483 0.13985 0.19866	140,100 115,400 89,690 128,100 184,900 93,070 29,090 17,900 13,060 9799 7702 11,420 79,660 293,500 455,500 533,500 555,900 536,400 477,500 401,900	4009 5437 7853 4367 2609 7404 44,536 89,333 132,165 185,996 238,782 116,596 5010 808 381 271 262 302 403 572	3.60 3.74 3.90 3.64 3.42 3.87 4.65 4.95 5.12 5.27 5.38 5.07 3.70 2.91 2.58 2.43 2.42 2.48 2.60 2.76	5.15 5.06 4.95 5.11 5.27 4.97 4.46 4.25 4.12 3.99 3.89 4.06 4.90 5.47 5.66 5.73 5.74 5.73 5.68 5.60
Measurement frequency at 10 kHz					
293 303 313 323 333 343 353 363 373 383 393 403 413 423 433 443 453 463 473 483	0.62512 0.85581 1.188 0.50394 0.3238 1.2082 7.2162 14.584 21.676 30.192 39.355 14.483 0.64942 0.12128 0.06145 0.04865 0.04694 0.05137 0.06369 0.08415	126,300 103,700 83,510 127,500 169,200 76,460 26,640 16,680 12,440 9304 7426 12,940 83,960 243,700 329,700 351,900 362,500 360,400 339,200 306,100	1800 2464 3420 1451 932 3478 20,774 41,985 62,402 86,918 113,297 41,694 1870 349 177 140 135 148 183 242	3.26 3.39 3.53 3.16 2.97 3.54 4.32 4.62 4.80 4.94 5.05 4.62 3.27 2.54 2.25 2.15 2.13 2.17 2.26 2.38	5.10 5.02 4.92 5.11 5.23 4.88 4.43 4.22 4.09 3.97 3.87 4.11 4.92 5.39 5.52 5.55 5.56 5.56 5.53 5.49

**Table 5 ijms-26-10409-t005:** (**a**)**.** Calculation of the band gap width (∆E) in the range of 293–313 K. (**b**)**.** Calculation of the band gap width (∆E) in the range of 343–363 K.

(a)	
T, K	lg R
293	4.41
313	4.22
**(b)**	
**T, K**	**lg R**
343	6.10
363	3.81

## Data Availability

The original contributions presented in the study are included in the article; further inquiries can be directed to the corresponding author.

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
