# Peer review of "Novel Spinel Li–Cr Nano-Ferrites: Structure, Morphology, and Electrical/Dielectric Properties"

_ijms, 2025, doi:10.3390/ijms262110409_

Round 1

Reviewer 1 Report

Comments and Suggestions for Authors

My review o the attached file

Author Response

Novel Spinel Li–Cr Nano-Ferrites: Structure, Morphology, and 2 Electrical/Dielectric Properties

Dear Reviewer,

We would like to express our sincere gratitude to you for carefully reviewing our manuscript entitled “Novel Spinel Li–Cr Nano-Ferrites: Structure, Morphology, and Electrical/Dielectric Properties.”

Your valuable comments and constructive suggestions have significantly contributed to improving the quality of our research and strengthening the scientific validity of the results. Based on your feedback, we have enhanced both the content and structure of the manuscript, raising its overall scientific level.

We sincerely thank you for dedicating your time and attention to our work. Your expertise and insightful feedback will guide our future scientific endeavors.

With best regards,

Madiyarova Altynai

Responses to the reviewer’s comments:

Question 1. “10.0 μm, 50.0 μm, and 500 μm” → changed to “10, 50, and 500 μm.”

Question 2. Figure 1 was enlarged and replaced with a clearer version; the XRD sample card number was also indicated.

Question 3. A reference to Figure 2 was added to the text.

Question 4. The “*” symbol in the word “Pycn.density” was removed.

Question 5. The information in lines 117–122 was already presented in Table 1; the repeated text was deleted.

Question 6. Figure 3 was enlarged, and axis labels were made clearer.

Question 7. Table 2 refers to crystallite sizes obtained at different temperatures; the average temperature value was used.

Question 8. The meaning of the different colors in Figure 4 was added to the text. The colors (black, red, and blue lines) mainly represent LiCr₃.₄Fe₁.₆O₈ ferrite samples calcined at different temperatures. The blue line corresponds to the sample synthesized at a low temperature, the red line to the medium temperature, and the black line to the high temperature. As the temperature increases, the peak shifts and intensities vary, indicating crystal structure improvement, stronger ionic bonding, or phase transformation.

Question 9.“5 μm, 30 μm, 100 μm, and 200 μm” → changed to “of 5, 30, 100, and 200 μm.”

Question 10. The element mapping in Figure 7 was obtained using the Thermo Scientific™ Axia™ ChemiSEM™ (Massachusetts, USA) instrument; this information was added to the text.

Question 11. Discussion of the EDX analysis was added.

Question 12. Figure 9 was discussed in the text.

Question 13. Figures 9–13 were described in the manuscript text.

Question 14. The symbols “B” and “L” were explained. Sample B (black square) represents the specimen with the lowest resistivity at low frequencies, while sample L (blue triangle) has one of the highest resistivities at low frequencies. The symbols B and L are used to distinguish between the two samples with different electrical properties in the graph.

Question 15. The redundant letter “D” in “electric displacement ?D” was removed.

Question 16. In Equation (10), “lg” was corrected to “log.”

Question 17. Line 330: “Table 1” → changed to “Table 3.”

Question 18. Line 333: “As show in Table 1 presents” → corrected to “As show in Table 3 presents.”

Question 19. Page 16, line 363: “Table 4. (a). Calculation” → corrected to “Table 5. (a). Calculation” & “lg R” → “log R.”

Question 20. Page 16, line 365: “Table 4. (b). Calculation” → corrected to “Table 5. (b). Calculation” & “lg R” → “log R.”

Question 21. Line 496: the journal volume number was added.

Question 22. Line 574: the journal volume number was added.

Question 23. Line 578: the journal volume number was added.

Reviewer 2 Report

Comments and Suggestions for Authors

I believe the manuscript should be revised. Firstly, the second and third sections can be combined into a single section titled "Results and Discussion." This section should not describe the equipment (Rigaku Miniflex, SPECTRUM TERS, LCR-800, etc.) used in the experiments, nor the experimental methods themselves. All of this should be moved to the "Materials and Methods" section. Secondly, the formulas and explanations of physical quantities should not be described in detail; it is sufficient to state that the Scherrer formula (or the Williamson-Hall equation) was used and provide a reference to the relevant literature. Thirdly, Figures 3 and 8 should be redone as they are of poor quality; furthermore, the table in Figure 8 is illegible. Fourthly, the readers of the journal would be interested if the authors linked the structure to the properties of the material. Otherwise, it turns out that the powder X-ray diffraction investigation of the structure is a separate part of the article, unrelated to the other studies, the spectroscopic study is another separate part, and the electrophysical research is also separate. Therefore, the manuscript resembles less a scientific publication and more a report on the work done.

Author Response

Novel Spinel Li–Cr Nano-Ferrites: Structure, Morphology, and 2 Electrical/Dielectric Properties

Dear Reviewer,

We would like to express our sincere gratitude to you for carefully reviewing our manuscript entitled “Novel Spinel Li–Cr Nano-Ferrites: Structure, Morphology, and Electrical/Dielectric Properties.”

Your valuable comments and constructive suggestions have significantly contributed to improving the quality of our research and strengthening the scientific validity of the results. Based on your feedback, we have enhanced both the content and structure of the manuscript, raising its overall scientific level.

We sincerely thank you for dedicating your time and attention to our work. Your expertise and insightful feedback will guide our future scientific endeavors.

With best regards,

Madiyarova Altynai

Responses to the reviewer’s comments:

Firstly, the second and third sections were combined into a single section titled “Results and Discussion.” In this section, the experimental equipment (Rigaku Miniflex, SPECTRUM TERS, LCR-800, etc.) and experimental methods were moved to the “Materials and Methods” section.

Secondly, detailed descriptions of formulas and explanations of physical quantities were deemed unnecessary; instead, it was mentioned that the Scherrer formula (or the Williamson–Hall equation) was used, with appropriate references provided.

Thirdly, Figures 3 and 8 were redrawn, and the improved versions were uploaded.

Fourthly, the relationship between the structure and the material properties has been demonstrated for the readers of the journal. The powder X-ray diffraction analysis was connected, as much as possible, with other sections of the manuscript. The spectroscopic and electrophysical studies were presented separately, as I believe it is more appropriate to describe each experimental method individually.

Round 2

Reviewer 2 Report

Comments and Suggestions for Authors

I think the manuscript is acceptable for publication.